# Long-distance electron transfer in a filamentous Gram-positive bacterium

Yonggang Yang[1,2,3], Zegao Wang[4,5], Cuifen Gan[1], Lasse Hyldgaard Klausen [4], Robin Bonné [6], Guannan Kong[1], Dizhou Luo[1], Mathijs Meert[6], Chunjie Zhu[1], Guoping Sun[1], Jun Guo[2], Yuxin Ma[7], Jesper Tataru Bjerg[8], Jean Manca [6], Meiying Xu [1,2,3✉], Lars Peter Nielsen [8] & Mingdong Dong [4✉]

Long-distance extracellular electron transfer has been observed in Gram-negative bacteria and plays roles in both natural and engineering processes. The electron transfer can be mediated by conductive protein appendages (in short unicellular bacteria such as *Geobacter* species) or by conductive cell envelopes (in filamentous multicellular cable bacteria). Here we show that *Lysinibacillus varians* GY32, a filamentous unicellular Gram-positive bacterium, is capable of bidirectional extracellular electron transfer. In microbial fuel cells, *L. varians* can form centimetre-range conductive cellular networks and, when grown on graphite electrodes, the cells can reach a remarkable length of 1.08 mm. Atomic force microscopy and microelectrode analyses suggest that the conductivity is linked to pili-like protein appendages. Our results show that long-distance electron transfer is not limited to Gram-negative bacteria.

[1] Institute of Microbiology, Guangdong Academy of Sciences, Guangzhou, China. [2] State Key Laboratory of Applied Microbiology Southern China, Guangzhou, China. [3] Guangdong Provincial Key Laboratory of Microbial Culture Collection and Application, Guangzhou, China. [4] Interdisciplinary Nanoscience Center (iNANO), Sino-Danish Center for Education and Research (SDC), Aarhus University, Aarhus, Denmark. [5] College of Materials Science and Engineering, Sichuan University, Chengdu, China. [6] X-LAB, Hasselt University, Diepenbeek, Belgium. [7] School of Life Sciences and Biopharmaceutics, Guangdong Pharmaceutical University, Guangzhou, China. [8] Center for Electromicrobiology, Aarhus University, Aarhus, Denmark. ✉email: xumy@gdim.cn; dong@inano.au.dk

Electron transfer is essential for energy generation and the metabolism of life. Bacteria are versatile in their ability to transfer electrons to various chemicals and to preserve energy. In addition to respiration with intracellular chemicals such as oxygen, sulfate, or nitrate, bacteria can also respire with externally accessed chemicals such as mineral particles (i.e., extracellular electron transfer, EET)[1–3]. Some single-celled bacteria are capable of long-distance electron transfer (LDET) to chemicals or other microbial cells at tens of micrometers distance[4–6] and multicellular cable bacteria transport electrons in the range of centimetres[7,8].

Two bacterial LDET strategies have been identified: conductive protein nanowires generated by relatively short, unicellular bacteria (e.g., *Geobacter* species) and conductive envelopes formed by filamentous, multicellular cable bacteria[4,8]. Conductive protein nanowires generated by *Geobacter* and several other microorganisms can transfer electrons over tens of micrometer[4–6]. These protein nanowires can form conductive networks and contribute to direct intercellular electron transfer in biofilms or aggregates composed of different microorganisms[9–11]. Cable bacteria usually form centimetre long filaments as they consist of thousands of cells end-to-end to couple sulfide oxidization in anoxic sediment and oxygen reduction at the sediment surface in aquatic systems[7,12,13]. The periplasmic fibers of cable bacteria have been shown to be conductive and can contribute to LDET[8]. The wide occurrence of bacterial LDET networks affects microbial communities and biogeochemical processes in natural and engineered environments across the earth. All reported bacteria capable of LDET so far have been Gram-negative, and although Gram-positive bacteria are ubiquitous and some are capable of EET, they have not been expected to evolve LDET, as they possess very different cell surface layers[3,14–16].

Here, we report an additional bacterial LDET strategy represented by *Lysinibacillus varians* GY32, which is capable of bidirectional EET. In microbial fuel cells (MFCs) strain GY32 can form centimetre-range conductive cellular networks composed of insulated filamentous cells with conductive nanowire-like appendages.

## Results

***L. varians* GY32 can form extremely long cells**. Strain GY32 is a unicellular filamentous Gram-positive bacterium isolated from freshwater sediment and contains multiple nucleoids in each cell[17,18]. Previously reported cell length of GY32 was up to about 500 μm with a uniform diameter (~0.5 μm) in nutrient-rich aerobic medium[17]. In this study, we found that GY32 could grow into a longer shape when anaerobically respiring with graphite electrodes as the sole electron acceptor in MFCs (Fig. 1a and Supplementary Fig. 1). The longest GY32 cell observed in MFCs was 1.08 mm (Fig. 1a), which is longer than *Thiomargarita namibiensis* (750 μm) and *Epulopiscium fishelsoni* (600 μm), the two biggest unicellular bacteria reported before[19]. Cell sections evidenced a single-cell structure of GY32 as no separation was observed inside the filamentous cells (Fig. 1b, c). It should be noted that some multicellular bacteria, such as cable bacteria, can grow to centimetre lengths as they form chains of thousands of cells connected end to end. However, each individual cable bacteria cell is less than 5 μm[3,13]. The filamentous shape of GY32 maintains a surface to volume ratio of 8 μm[−1], which is similar to that of rod-shape bacteria such as *Escherichia coli* (4.5 μm[−1]) and much higher than that of big coccus *T. namibiensis* (0.004 μm[−1]) and rod-shape *E. fishelsoni* (0.05 μm[−1])[19]. A higher surface to volume ratio supports a faster rate of nutrient and waste exchange per unit of cell volume and thus a faster growth rate.

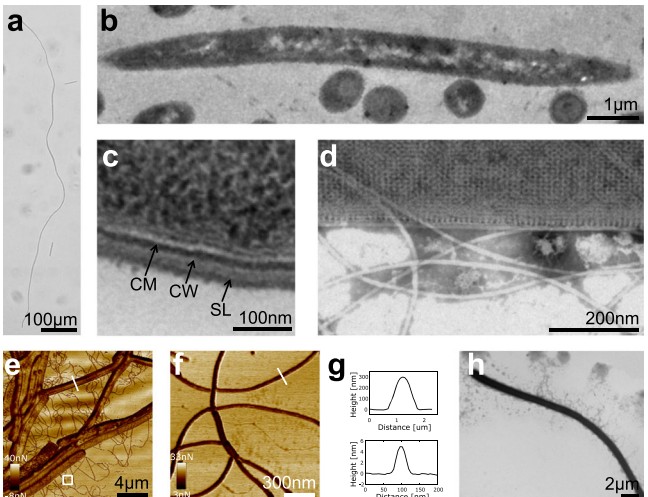

**Fig. 1 Structures of *L. varians* GY32 cell, cell wall, and nanowire-like appendages. a** The longest GY32 cell. **b**, **c** Longitudinal and cross-sections of GY32 observed with TEM. CM cytoplasm membrane, CW cell wall, SL S-layer. **d** Structure of the surface layer and appendages observed with TEM. **e**, **f** AFM adhesion maps of GY32 cells and appendages. F is a zoom-in of the white square in (**e**). **g** AFM height profiles of the cell (top) and appendage (below). **h** Representative cell of strain GY32 observed with TEM.

Most Gram-positive bacteria have multilayer peptidoglycan as the major scaffolding structure in the cell wall. In GY32 we furthermore observe two-dimensional crystal layers on the surface (Fig. 1d), indicating a protective S-layer structure on top of the peptidoglycan. The sectioning results (Fig. 1c) suggested width of 20–30 nm of the cell wall, which is comparable to some other Gram-positive bacteria and much thicker than that of the Gram-negative bacteria (2–7 nm). In addition to contributing to the mechanical properties of the cell body, the thick cell wall is generally considered to be a limiting factor for the extracellular electron shuttling capacity of Gram-positive bacteria[16]. Outside the cell wall, pili-like or nanowire-like appendages sticking to the cells were observed by transmission electron microscope (TEM) and atomic force microscopy (AFM) (Fig. 1d–h). The diameter of individual appendages was 6–8 nm as measured by TEM and 4–6 nm as measured from the height profile in AFM (Fig. 1g). The appendages often bind together and tangle, and the length of an individual appendage could be more than 10 μm. The size and structure of the appendages are similar to both type IV pilin and multi-heme c-type cytochrome nanowires of *Geobacter sulfurreducens*[4,20].

***L. varians* GY32 can transfer electrons to graphite electrodes in both sediment and liquid environments**. To test the possible EET capacity of strain GY32, sediment MFCs (SMFCs) were assembled in which the anode (graphite plate) can serve as a solid electron acceptor for microorganisms and the electricity can be used to evaluate the microbial EET rate in sediment (Fig. 2a, Supplementary Fig. 2)[21]. In the SMFCs containing sediment from the river where GY32 was isolated[17], the current density increased by 75.2 ± 7.1% after artificial inoculation of GY32 compared to those without GY32 supplement (2.8 ± 0.3 vs. 1.6 ± 0.1 μA/cm², Fig. 2b). A stable current could be maintained for over 10 days, indicating a relatively stable role of GY32 in the sedimentary microbial community. We also assembled SMFCs with sterilized sediments, where GY32 was supplemented to function as the only bacterium. This SMFC generated a current density of 0.9 ± 0.04 μA/cm², which was fivefold

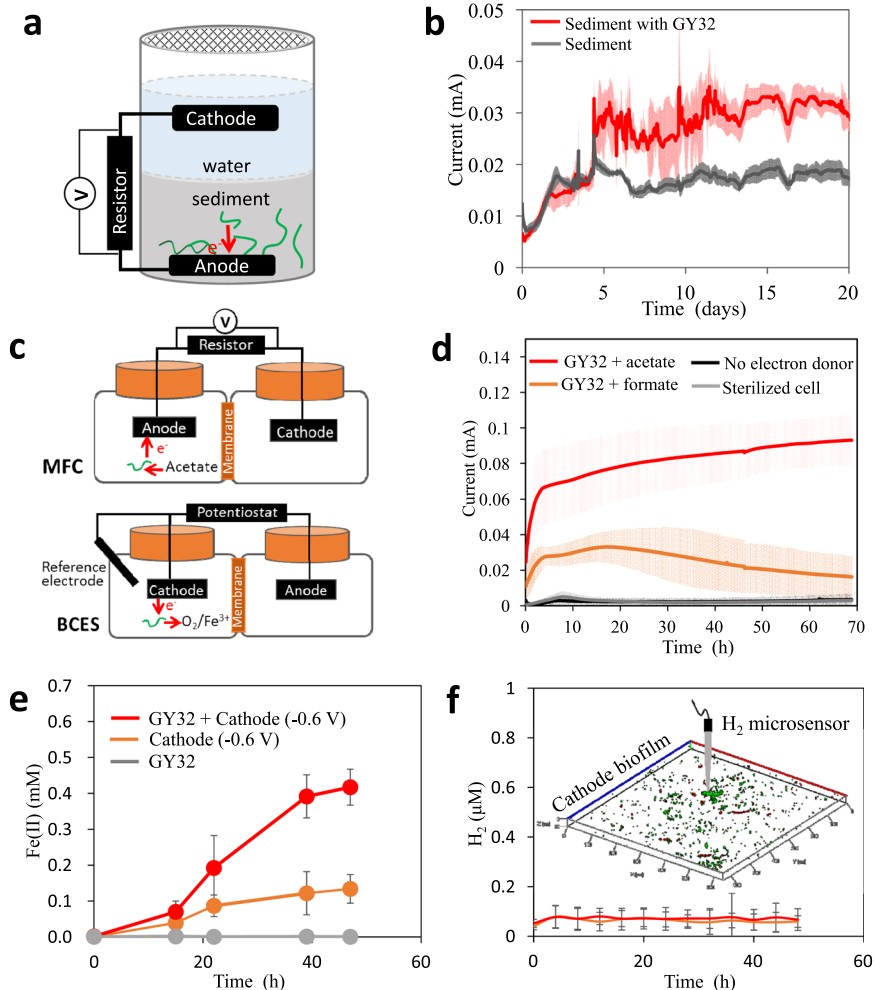

**Fig. 2 Bidirectional EET between GY32 and electrodes. a** Schematic of GY32 electricity generation in SMFCs. **b** Electricity generation by inoculating GY32 in SMFCs. **c** Schematic of GY32 EET in MFCs with liquid medium (upper reactor) and in BCESs (lower reactor). **d** Electricity generation by GY32 in MFCs with liquid medium. **e** Fe(III) reduction by GY32 with a cathode (− 0.6 V vs. SHE) as the sole electron donor in BCESs. **f** $H_2$ concentration in the BCESs with or without GY32, inset shows the GY32 biofilms on the cathode. $n = 3$ for each experiment, plots show mean and standard deviation. Source data are provided as a Source Data file.

higher than that of the background electricity of the sterilized sediments (Supplementary Fig. 2). Live/dead bacteria staining showed that the filamentous GY32 were still alive in the sediments on day 20 after inoculation, and the cell density increased toward the anode surface (Supplementary Fig. 2). These results suggest that strain GY32 can grow in sediments through EET to solid electron acceptors as well as contribute to microbial EET processes in natural environments.

In MFCs containing defined liquid medium (Fig. 2c), GY32 could use acetate or formate as an electron donor for the reduction of graphite anode. In acetate-fueled MFCs, the maximum current density was $7.5 \pm 1.2\,\mu A/cm^2$ after being incubated for 69 h (Fig. 2d). Moreover, the biomass of planktonic cells increased simultaneously with the acetate consumption and electricity generation (Supplementary Fig. 2E), suggesting that GY32 respired and grew with the anode as the sole electron acceptor. When respiring with anodes polarized at 0.4 V (vs. standard hydrogen electrode, SHE), strain GY32 generated a maximum current density of $11.7 \pm 1.1\,\mu A/cm^2$ (Supplementary Fig. 2F). The current densities of MFCs catalyzed by *L. varians* GY32 were comparable with that of *Shewanella oneidensis* MR-1 using a graphite anode. Cyclic voltammetry curves of anodic GY32 biofilms showed an oxidative peak at 0.09 V and a reductive peak

at −0.08 V vs. SHE (Supplementary Fig. 2G) under non-turnover condition. A similar cyclic voltammetry profile with higher peak-current was observed in the presence of acetate as an electron donor (turnover condition), indicating redox species generated within biofilms.

We further tested the capability of *L. varians* GY32 using electrode as electron donor and Fe(III) citrate as electron acceptor in a biocathode electrochemical system (BCES, Fig. 2c). With a graphite cathode polarized at −0.6 V (vs. SHE), GY32 showed a Fe(III) citrate reduction rate of $8.9 \pm 1.1\,\mu M/h$, while the polarized graphite cathode without GY32 showed a Fe(III) reduction rate of $2.9 \pm 0.5\,\mu M/h$, and GY32 without the polarized cathode did not reduce Fe(III) (Fig. 2e). The possibility that $H_2$ was generated on the negatively polarized electrode and serves as an electron donor to Fe(III) reduction could be ruled out[22,23] because no $H_2$ was accumulated in the cathode chamber ($H_2$ concentration maintained lower than 0.1 μM, Fig. 2f) and *L. varians* GY32 cannot reduce Fe(III) even in the presence of $H_2$ (Supplementary Fig. 3). These results suggested that GY32 used the cathode as the sole electron donor to reduce Fe(III) in the BCESs. Moreover, *L. varians* GY32 could also use a polarized electrode (0.1 V vs. SHE) as an electron donor to reduce dissolved oxygen (Supplementary Fig. 4). Integrating with the electricity generating

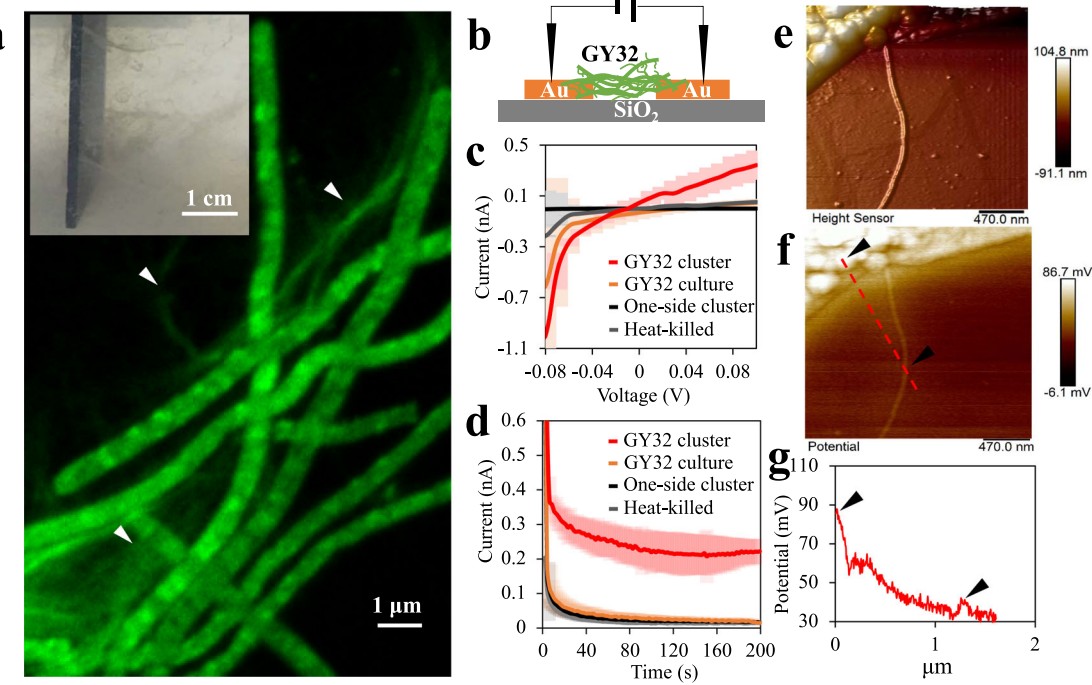

**Fig. 3 Conductive cellular networks of strain GY32. a** GY32 networks comprising filamentous cells and nanowire-like appendages stained by Nano-Orange. Inset shows the in situ networks around anode in an MFC, triangles indicate appendages. **b** Schematic of conductance measurement of the hydrated GY32 cell cluster. **c** Current–voltage profiles of live GY32 cell clusters and control samples ($n = 18$ independent bacterial cultures). **d** Current–time profiles of live GY32 cell clusters and control samples (the voltage is 0.1 V, $n = 18$ independent bacterial cultures). **e** 3D-topography image of dehydrated GY32 and an appendage. **f** Surface potential image according to (**e**). **g** Representative surface potential profile along the red line in (**f**), triangles indicate the locations of cell body and appendage. Data in (**c**) and (**d**) are plotted as mean and standard deviation. Source data are provided as a Source Data file.

capabilities, our results showed that GY32 could behave as either electron donor or electron acceptor to extracellular redox particles. Such bidirectional EET capability of GY32 implies a possibility that GY32 could perform interspecies electron transfer with another bacterial cell. Moreover, since electron mediator synthesis would be hampered by the lack of a carbon source in the BCESs, it is more likely that GY32 received electrons from the cathode mainly by direct contact between the cells and the cathode surface.

***L. varians* and its appendages form conductive cellular networks**. When transferring electrons to anodes in MFCs, the filamentous strain GY32 preferred to cluster and form network-like structures surrounding the anode surface (Fig. 3a, Supplementary Fig. 5). These networks could extend several centimetres from the anode surface into the liquid medium. The cell clusters developed unevenly on the anode surface, and most cells maintained the filamentous structure and high viability during electricity generation (Supplementary Fig. 5B). Figure 3a also showed that nanowire-like appendages along the filamentous cells formed subnetworks bridging different cells in the clusters. After being stained by Nano-Orange, a protein-specific fluorescent dye that has been used to visualize bacterial nanowires of *Geobacter* and *Shewanella* species[24,25], significant fluorescence was observed along with the cell-attached appendages, indicating that these are composed of protein (Fig. 3a).

The cell clusters from the cellular networks, consisting of GY32 cells and their appendages, were picked and used to connect two prefabricated gold electrodes with a 0.1 mm insulating (SiO$_2$) gap (Fig. 3b). When the voltage between the two electrodes was swept from −0.1 to 0.1 V, the current increased with the voltage (Fig. 3c).

When a fixed voltage of 0.1 V was applied, stable currents of 0.25 ± 0.06 nA were observed (Fig. 3d), suggesting an electronic conductance capability of the cell clusters. The calculated conductivity of GY32 cell clusters ranged from 0.1 to 0.2 mS cm$^{-1}$ ($n = 18$), depending on the structure and growth stage of GY32 cell clusters. This conductivity was in the range of the conductivity of *G. sulfurreducens* biofilms and mixed-species biofilms or microbial aggregates (from several to thousands of μS cm$^{-1}$)[26–28]. In contrast, control samples including (a) clusters of heat-killed GY32 cells, (b) clusters not connecting the gold electrodes, (c) culture supernatant showed much smaller responses to voltage. Moreover, the current obtained for the control samples at 0.1 V was one order of magnitude lower (0.01–0.04 nA) than that of the live GY32 cell clusters (Fig. 3c, d). The conductive GY32 cell clusters suggested that centimetre-range cellular networks around the anode were conductive. The cell clusters conductivity decreased when dried in air and could be recovered when rehydrated with deionized water, which was similar to *G. sulfurreducens* biofilms and consistent with a redox conductivity[26,29]. Moreover, electrochemical gating measurements of biofilms grown on interdigitated microelectrode arrays (IMAs) showed a peak conductivity of 0.25 ± 0.1 mS cm$^{-1}$ at 0.05 V (vs. SHE) when the gate potential increased from −0.4 to 0.2 V (vs. SHE, Supplementary Fig. 6), which also suggested that the LDET of *L. varians* GY32 cellular networks is a redox process[29].

Kelvin probe force microscopy (KPFM) was then used to further understand the electrical properties of nanostructures in GY32 cell clusters. The samples were prepared on highly conductive gold-coated mica substrates and measurements were conducted using a two-pass lift mode. The first pass was used to probe the topography of the sample (Fig. 3e), and in the second pass, the probe was traced along with the sample at a certain height. A potential is applied between the probe and sample

during the second pass to nullify interaction between the probe and sample in a frequency modulated feedback. The contact potential difference (also named surface potential) between the conductive tip and the sample was recorded during the second scan and presented with the gold background set to 0 (Fig. 3f). The potential difference between the nanowire-like appendages and gold substrate was much smaller ($5 \pm 2$ mV) than that between dried GY32 cell and gold substrate ($53 \pm 11$ mV) (Fig. 3f, g). Unlike conductivity tests, KPFM does not produce a current flow but rather measures the long-ranged electrostatic interactions in the sample-probe capacitor set-up. The small potential difference between nanowire and gold showed a similar work function and could indicate that the appendages are much more conductive than the cell envelope of GY32.

**Microelectrode tests suggested conductivity of GY32 appendages.** The filamentous cells and their appendages are the main components in GY32 cellular networks. Based on the reported conductive cell envelope of cable bacteria and conductive nanowires of *Geobacter* species[4,8], the conductivity of both cell envelope and appendages of GY32 should be further evaluated. When we used a single GY32 cell (without appendages connecting the electrodes) to connect fabricated electrode arrays (Fig. 4a, b), no significant current was detected within a direct voltage of 0.1 V. Moreover, no obvious current was obtained when the voltage ranged from −0.1 to 0.1 V (Fig. 4c), regardless of using dried, wet, live or heat-killed cells, indicating that the GY32 cell envelope is not conductive. On the other hand, when electrodes on IMAs were connected by nanowire-like appendages

extracted from GY32 cells (Fig. 4d, e), significant current ($24.8 \pm 4.0$ pA, $n = 8$) was observed when a voltage of 0.1 V was applied between the electrodes, while the buffer showed only a current of $2.0 \pm 0.5$ pA. Similar to the GY32 cell clusters, the conductivity of appendages decreased when dried in air. These results strongly suggest that the appendages are conductive and can be responsible for the conductivity of the cell networks, although further study is needed to elucidate the chemical composition and electrical behavior of an individual appendage.

**Possible components participating in the LDET of *L. varians* GY32.** Dominant roles of Gram-positive bacteria have been evidenced in many microbial communities capable of reducing metal oxides or electrodes. EET strategies generally involve *c*-type cytochromes or electron shuttles, and EET has been observed for the Gram-positive bacteria *Enterococcus faecalis* and *Listeria monocytogenes* using flavins as electron shuttles[16,30–33]. However, no available information was reported about nanowire-like appendages of Gram-positive bacteria in EET or LDET.

The nanowire-like appendages of GY32 are too thin to be cell membrane extensions (10–150 nm) as found in *S. oneidensis* MR-1 or flagella (15–20 nm)[19,24]. The appendages of GY32 have a similar size to the conductive type IV pilus of some Gram-negative bacteria (i.e., e-pili). In the genome of *L. varians* GY32 (accession number NZ_CP006837.1, https://www.ncbi.nlm.nih.gov/nuccore/NZ_CP006837.1), gene *T479_RS14015* (https://www.ncbi.nlm.nih.gov/nuccore/754147054) is predicted to encode ComGD, a putative type IV pilin. The high density of aromatic amino acids is considered to be a key property in

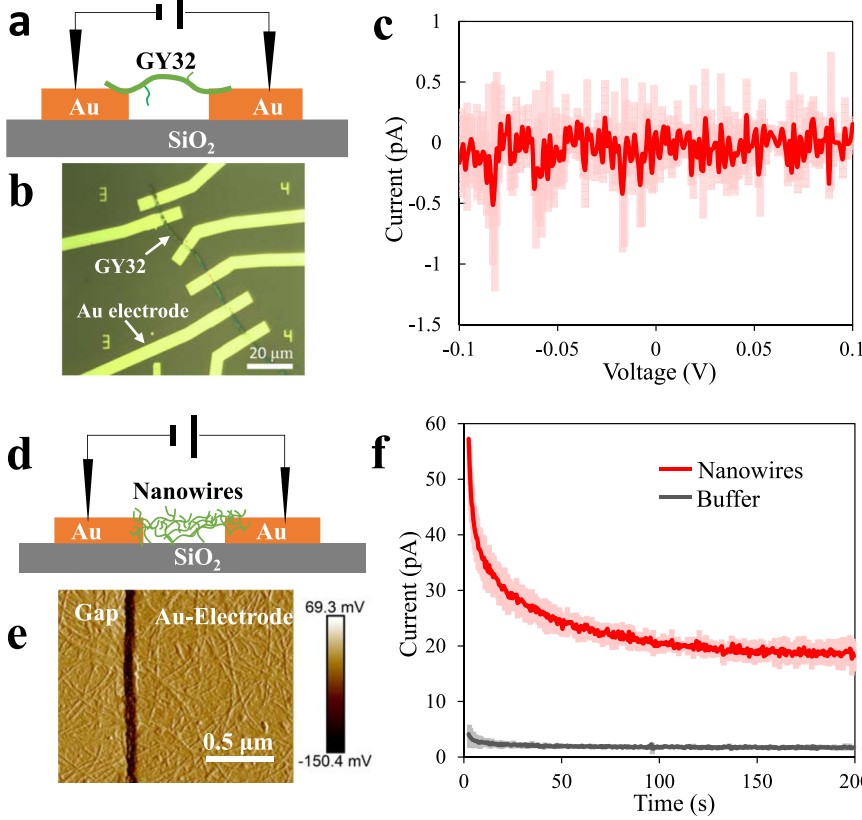

**Fig. 4 Conductivity measurements of the GY32 cell envelope and nanowire-like appendages by microelectrodes. a** Schematic of the single-cell measurement; **b** single cell on an electrode array observed under a light microscope; **c** current–voltage curve of a single GY32 cell ($n = 5$, each from independent bacterial culture); **d** schematic of the appendage measurement; **e** appendage network on an electrode array observed by AFM, the dark line indicates a 100 nm height difference between the gap and electrode; **f** current–time curve of the hydrated appendage products (the voltage is 0.1 V, $n = 8$ independent bacterial cultures). Plots show mean and standard deviation. Source data are provided as a Source Data file.

conductive type IV pili and archaellum[6,34,35]. The percentage of aromatic amino acids in GY32-ComGD is 13.8% (19/137 amino acids, Supplementary Fig. 7), which is comparable or even higher than that of tested conductive pili[6]. The largest gap between those aromatic amino acids is 24 amino acids, also smaller than the suggested upper limit of 35 amino acids. In addition, three key amino acids are usually considered in e-pili, a phenylalanine (F1) initiating the α-helix domain, a glutamic acid residue in position +5 (E5), and a tyrosine at position +57 (Y57, catalyzing electron transfer to extracellular electron acceptors)[36,37]. GY32-ComGD has E5 and Y57 while the F1 was replaced with an aromatic amino acid tyrosine (Supplementary Fig. 7). We speculate that the conductive appendages of GY32 may be composed of ComGD and may have a similar conductance mechanism to the e-pili of *Geobacter* species[26,34].

In addition to pili, linear polymers of *c*-type cytochromes can form conductive appendages[20,38]. GY32 has six putative *c*-type cytochrome genes (containing heme-motif CXXCH). Among them, *T479_RS14495* (https://www.ncbi.nlm.nih.gov/nuccore/754147054), *T479_RS06030* (https://www.ncbi.nlm.nih.gov/nuccore/754147054), *T479_RS08690* (https://www.ncbi.nlm.nih.gov/nuccore/754147054), and *T479_RS20980* (https://www.ncbi.nlm.nih.gov/nuccore/754147054) showed significantly higher transcription level in MFCs compared to aerobic growth (transcriptome data accession number: GSE165753, and summarized in Supplementary Data 1). Consistently, cytochrome-specific staining after protein electrophoresis showed that cells grown in MFCs have a higher concentration of c-type cytochromes (Supplementary Fig. 8). These results suggest that c-type cytochromes might play a role in the EET of GY32. However, whether and how these *c*-type cytochromes participate in the LDET and EET by GY32 needs more studies.

Bacteria can generate a range of appendages with a distinct function, structure, and molecular composition, including flagella[19], pili[34], cell membrane extensions[24], c-type cytochromes polymers[20,38], pseudopilins, and other unknown nanowires[39]. Although important environmental roles and promising applications of bacterial protein nanowires have been reported, the compositions and mechanisms of many of these nanostructures are still unclear[34,39–41].

## Discussion

Here, we have shown that the unicellular bacterium *L. varians* GY32 is capable of bidirectional EET and can form centimetre-scale conductive cellular networks, growing into remarkably long cells when anaerobically respiring with graphite electrodes as the sole electron acceptor.

Gram-positive bacteria are ubiquitous in various environments and can be dominant in some iron- or electrode-reducing microbial communities[16,42]. Although several of them have been demonstrated to perform EET via outer membrane cytochromes or electron mediators, current knowledge on the strategies of Gram-positive bacteria to participate in microbial EET and LDET networks are much less compared to these of Gram-negative bacteria[3,16,30–33]. Our results show that the filamentous Gram-positive bacterium *L. varians* GY32 is capable of bidirectional EET and can form centimetre-scale conductive cellular networks. KPFM, electrode array tests, and gene expression analyses suggest that nanowire-like appendages, and possibly *c*-type cytochromes, may be key components in the EET and LDET of the conductive cellular networks. However, the detailed composition and mechanism of GY32 electric networks need to be elucidated.

More and more evidence suggests a wide distribution of microbial LDET in natural and engineering environments[3–5,7,11,27]. Two types of microbial LDET networks have been intensively studied.

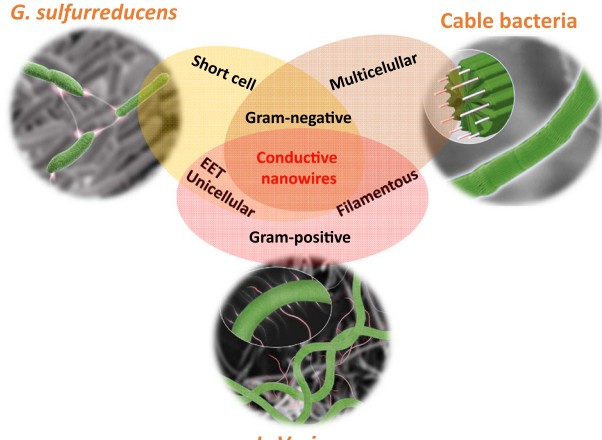

**Fig. 5 A comparison among the conductive networks of *Geobacter*, cable bacteria, and *L. varians* GY32.** *Geobacter* can form conductive networks (i.e., biofilms) comprising short unicellular bacteria and extracellular conductive protein nanowires[4]. Filamentous multicellular cable bacteria conduct electricity through intercellular nanowires in the bacterial envelope[8]. *L. varians* GY32 combines a filamentous shape with extracellular nanowire-like appendages to form conductive networks.

Firstly, short cells (e.g., *Geobacter* and *Shewanella*) integrated with extracellular nanowires. These LDET networks depend largely on the biofilm formation on particle surfaces and have a limited thickness (generally ≤ 0.1 mm)[37]. Secondly, filamentous cable bacteria with serially connected cells and intracellular nanowires[7,8]. Cable bacteria LDET networks can span centimetre distances but exist mainly at sediment–water interfaces. The LDET networks of Gram-positive *L. varians* GY32 comprise filamentous, unicellular cells and extracellular nanowire-like appendages, resembling a combination of the two LDET network modes of Gram-negative bacteria (Fig. 5). Centimetre-long cell networks of GY32 can form around the solid electron acceptors (Fig. 3a). Given that GY32 is a facultative anaerobic bacterium, it might potentially form conductive networks in both oxic and anoxic environments.

## Methods

**Bacterial culture**. *L. varians* GY32 was isolated from heavy metals-polluted sediment and preserved in our laboratory[17]. Strain GY32 was cultivated aerobically in Luria–Bertani broth (LB) for 12 h (30 °C, 120 rpm) in flasks. The LB (1 L) contained 5 g yeast extract (LP0021, Oxoid, United Kingdom), 10 g Tryptone (LP0042, Oxoid, United Kingdom), and 5 g NaCl. Before being used for inoculation to MFCs or further tests, the LB-cultivated bacteria cells were washed in sterilized phosphate buffered saline (PBS) or deionized water for at least three times by centrifugation at 8000×$g$ for 5 min. PBS used here (pH 7.2, 1 L) contained 8 g NaCl (C111545, Aladdin, China), 0.2 g KCl (P112133, Aladdin, China), 3.63 g Na$_2$HPO$_4$·12H$_2$O (S112623, Aladdin, China), 0.24 g KH$_2$PO$_4$ (P104071, Aladdin, China), the pH was adjusted by HCl (10011008, Hushi, China).

**Nanowire appendage preparation**. Nanowire-like appendages were extracted following the methods reported before[20]. The final nanowire preparations were collected in 2 ml dialyzed ethanolamine buffer and stored under 4 °C before microscope and conductivity tests. The ammonium sulfate (A801012) and ethanolamine (E808764) used in this experiment were bought from Shanghai Macklin Biochemical Co., Ltd., China.

**MFC assembly and operation**. For MFCs containing liquid medium, three dual-chamber MFCs were assembled and operated as reported previously with minor modifications[43]. Briefly, plain graphite plates (2 × 3 × 0.1 cm, Koboi, China) were used for both anodes and cathodes. Anodic medium (1 L) contained 12.8 g/L of Na$_2$HPO$_4$ (S818118, Macklin, China), 3 g/L of KH$_2$PO$_4$ (P815662, Macklin, China), 0.5 g/L of NaCl (S805275, Macklin, China), 1.0 g/L of NH$_4$Cl (A801304, Macklin, China) and 5 mM of formate (S164504, Aladdin, China) or acetate (S118649, Aladdin, China) as the sole electron donor. PBS supplemented with 50 mM potassium ferricyanide (C01512028, Macklin, China) was used as catholyte. The anode culture was inoculated with the PBS-washed *L. varians* cells with an initial OD$_{600}$ of 0.02. The anode chambers of the MFCs were bubbled with pure N$_2$ until

the dissolved oxygen in the medium was below 0.3 μM with an oxygen micro-electrode (detection limit: 0.3 μM, OX-14125, Unisense, Denmark). Titanium wire with a 1000 ohm resistor was used to connect the cathode and anode in each MFC and the voltage of the resistor was recorded with a multimeter (Keithley 2700, USA) to calculate the current. MFCs were operated under 30 °C. The MFC setup is illustrated in Fig. 2c. Anodic bacterial growth was determined by quantifying the total protein concentration of anode culture with a Bradford protein assay kit (C503031, Sangon Biotech, China). Electron donor consumptions were evaluated periodically during MFC operation with a high performance liquid chromato-graphy system (1525, Waters, USA) equipped with a Waters 2487 detector[44]. A Zorbax SB-Aq (880975-914, Agilent, USA) column was used at 35 °C and operated with a mobile phase containing 2% $CH_3OH$ and 98% of 0.01 M $K_2HPO_4$ (pH 2.8). The flow rate was 0.6 mL/min and the detection wavelength was 210 nm.

To further test the EET capability of *L. varians* GY32 to a polarized electrode, the same MFC reactors and electrolyte were prepared except for that the external resistors were replaced with an electrochemical workstation (CHI 1040C, China) and an Ag/AgCl reference electrode (1038, Gaoss Union, China) was placed in each anode chamber. The anodes were polarized at 0.203 V vs. Ag/AgCl reference electrodes (i.e., 0.4 V vs. SHE). The current between anode and cathode was recorded for every 100 s under 30 °C by the electrochemical workstation.

SMFCs were assembled with a method described in our previous report[45]. Sediments sampled from Lianjiang (Guiyu, China, where strain GY32 was isolated, N23°17′39.32″, E116°21′1.93″) were treated with a 0.5 cm sieve and stored under 10 °C before use. To obtain sterilized sediment, some of the sieved sediment was autoclaved (120 °C 20 min) for three times (once every 2 days). Totally, 50 mL of sterilized or non-sterilized sediment was inoculated with PBS-washed strain GY32 (to a final density of approximately 1000 cells/mL), and then homogenized and covered with 20 mL of sterilized tap water in beakers. Plain graphite felts ($2 \times 0.4 \times 2$ cm, Koboi, China) were used for anodes in sediment and cathodes in water. To exclude ambient microorganisms, the beakers were covered with eight layers of gauze. Titanium wires with a resistor of 1000 ohm were used to connect the anodes and cathodes in all SMFCs. The current through the resistor was monitored by a multimeter (2700, Keithley, USA). Fifteen SMFCs were operated here, including three SMFCs containing sterilized sediments, three SMFCs containing sterilized sediments inoculated with strain GY32, three SMFCs containing sterilized sediments inoculated with strain GY32 and 10 mM of formate, three SMFCs containing unsterilized sediments, and three SMFCs containing unsterilized sediments inoculated with strain GY32. All SMFCs were operated at room temperature ($25 \pm 3$ °C). The setup is illustrated in Fig. 2a and a photograph is shown in Supplementary Fig. 2A.

**BCESs assembly and operation**. Three dual-chamber BCESs were assembled as previously reported[43]. Graphite plates ($2 \times 3 \times 0.1$ cm, Koboi, China) connected to titanium wires were used for both anodes and cathodes. For cathodic Fe(III) reduction, both cathode and anode chamber contained 100 mL sterilized PBS. The cathode chamber was supplemented with GY32 (washed for three times with sterilized PBS, initial $0D_{600} = 0.05$) and 2 mM of ferric citrate (F110791, Aladdin, China). The cathode potential was poised at −0.6 V (vs. SHE) by using an electrochemical workstation (CHI 1040C, China) and an Ag/AgCl reference electrode. The cathode chambers were bubbled with pure $N_2$ until the oxygen in the PBS was below 0.3 μM as measured with an oxygen microelectrode (detection limit: 0.3 μM, OX-14125, Unisense, Denmark) and then sealed with butyl rubbers and plastic caps. BCESs with either unpolarized cathodes or GY32-free catholyte were oper-ated as controls. The $Fe^{2+}$ concentration was tested by ferrozine assay[46] and $H_2$ concentration at the surface of the cathode was monitored with a $H_2$ microelec-trode ($H_2$-25-7343, Unisense, Denmark). Similarly, three dual-chamber BCESs were assembled for cathodic oxygen reduction. The catholyte was 100 mL sterilized PBS and the dissolved oxygen was elevated to $233.8 \pm 2.7$ μM by bubbling with air. The initial $OD_{600}$ of PBS-washed GY32 cells in catholyte was 0.05. Each anode chamber was supplemented with 100 mL sterilized PBS. The cathode potential was poised at 0.1 V (vs. SHE) by using an electrochemical workstation (CHI 1040C, China) and an Ag/AgCl reference electrode. The cathode chambers were sealed with butyl rubber gaskets and plastic caps. Triplicate BCESs with either unpolar-ized cathodes or GY32-free catholyte were operated as controls. The dissolved oxygen in the cathode chamber was tested using the oxygen microelectrode (OX-14125, Unisense, Denmark). All BCESs were tested in an anaerobic workstation (AEP, Electrotek, United Kingdom) under 30 °C. The setup is illustrated in Fig. 2c.

**Cyclic voltammetry and electrochemical gate measurements**. After the tests of the current generation by GY32 with 0.4 V-polarized anodes, cyclic voltammetry profiles of GY32 biofilms on working electrodes were measured under turnover condition (with acetate as electron donor). Then the anodic cultures in the three MFCs were replaced with anaerobic ($N_2$-bubbled) fresh PBS while the anodic GY32 biofilms and Ag/AgCl reference electrodes were maintained and non-turnover cyclic voltammetry profiles were recorded. During cyclic voltammetry measure-ments, the anode potential varied between −0.4 and 0.3 V (vs. SHE) with a scan rate of 2 mV/s, data of the third cycle were presented. The cyclic voltammetry of bare electrodes in anaerobic PBS was also tested to evaluate the background current.

Electrochemical gating measurements of GY32 biofilms were conducted using a bipotentiostat model of Autolab PGSTART302N electrochemical workstation (Metrohm, Switzerland), as described before[47]. Biofilms of *L. varians* GY32 were grown on IMAs (Yuxin, China) serving as anodes in MFCs. The IMAs consisted of 50 parallel gold rectangular bands, each 2 mm long × 20 μm wide × 100 nm thick, with 20 μm gaps patterned onto a $SiO_2$ substrate. The two ends of IMAs were connected to Ti-wires via silver pastes (Electrolub, United Kingdom). Biofilms grown on IMAs in aerobic LB medium (30 °C for 24 h) were observed under a microscope to ensure the separated electrode arrays were connected by biofilms. Biofilms-connected IMAs and Pt-wires were then used as working electrode and counter electrode, respectively, in glass bottles containing 100 mL of anaerobic PBS as electrolyte and Ag/AgCl reference electrodes. The two arrays on IMAs were used as source and drain electrodes, respectively. While the source potential was increased from −0.4 to 0.2 V (vs. SHE) step wisely, the drain potential was increased simultaneously and maintained a 0.01 V voltage ($V_{sd}$) between the source and the drain electrode. The increment between every two steps was 0.05 V and the maintaining time for each step was 300 s to allow the transient charging current to decay. The current differences ($I_{sd}$) between the drain and source currents were calculated and divided by 2 ($I_{sd}$) and then used to calculate the conductivity ($\sigma$) of biofilms with the formula $I_{sd} = \sigma S V_{sd}$[47], where $S$, the IMAs geometric factor, was calculated to be 6.6 cm for the IMAs used here. It should be noted that the calculated conductivity can be considered as a minimum value as the calculation assumed that the IMAs were fully covered by biofilms, which is different from the unevenly distributed biofilms on IMAs (Supplementary Fig. 6). Three biological repeats were included in this experiment.

**Atomic force microscopy**. Bacterial cells or the nanowire filaments were immo-bilized on glass slides and loaded on a Bioscope Resolve AFM (Bruker, USA). The AFM was operated in PeakForce QNM mode and using SCANASYST-AIR probes (nominal spring constant 0.4 N/m, nominal tip radius 2 nm) (Bruker AFM Probes, USA). All images were further processed and analyzed by Nanoscope Analysis 1.9 (Bruker, USA). KPFM mode was used to measure the surface potential of GY32 cells and the nanowire appendages with electrically conductive SCM-PIT probes (Bruker AFM Probes, USA). The bacteria samples were loaded on gold-coated mica substrates or highly oriented pyrolytic graphites and then washed with deionized water five times. After being air-dried, the samples were scanned with a lift scan height of 50 nm and a drive amplitude of 2 V. The AFM investigation was per-formed for three independent cultures of *L. varians* GY32 all showing similar results.

**Direct current (DC) conductivity tests via electrode arrays**. The conductivity of cell clusters and nanowires was tested under fully hydrated conditions[26]. To test the conductivity of the cell clusters, cell clusters ($n = 18$ different bacterial cultures) were picked from the MFC cultures with a glass hook and then washed with deionized water five times. The clusters were then loaded on a prefabricated electrode array with 1 mm wide gold electrodes separated by 0.1 mm insulating $SiO_2$-gaps. The DC measurements through the fully hydrated cell clusters were performed with a probe station equipped with a Keithley 2614B source meter in an ambient environment (25 °C). In addition to cell clusters, control samples including uncentrifuged bacteria culture, culture supernatant, heat-killed cell clusters, and deionized water were also tested. A voltage, typically ranged from −0.1 to 0.1 V, was applied between the two gold electrodes to test the $I$–$V$ profiles of different samples with a scan rate of 10 mV/s. $I$–$t$ profiles of the samples at a voltage of 0.1 V maintained for 200 s were also tested. The conductivity $\sigma$ of a typical cellular cluster was calculated using the formula $\sigma = l/(AR)$[8], where $l$ is the conduction length of 100 μm and $A$ is the cross-section of about 40 μm$^2$ (cluster width of 40 μm with an average height of 1 μm, determined by AFM). The resis-tance $R$ is found from a current/voltage-measurement (−0.1 to 0.1 V, scan rate 10 mV/s) to be about $0.25 \times 10^9$ ohms, resulting in conductivity of 0.1 mS/cm. This can be seen as a minimum value for the conductivity since it assumes the whole cellular network to be responsible for the conduction.

The same IMAs used in the above electrochemical gate measurements were also used to test the conductivity of nanowire appendages extracted from GY32 culture. One microlitre buffer containing nanowire appendages ($n = 8$), or ethanolamine buffer without nanowire appendages as a control, was loaded on an electrode array. The electrode array was then observed with AFM to confirm the connection of electrodes by nanowires (Fig. 4e). The $I$–$V$ (from −0.1 to 0.1 V, scan rate 1 mV/s) and $I$–$t$ (0.1 V maintained for 200 s) profiles of the electrode array connected by nanowires were tested.

To test the conductivity of the individual cells, 2 μL of GY32 culture, after being diluted by 1000-fold with deionized water, were loaded on a laboratory fabricated electrode array (Fig. 4b). Cells ($n = 5$) connecting two or more electrodes were located under an optical microscope and the $I$–$V$ profile between the electrodes connected by these cells were tested (from −0.1 to 0.1 V, scan rate 2 mV/s) with the probe station equipped with a Keithley 2614B source meter in the ambient environment (25 °C).

**Confocal laser scanning microscopy**. PBS-washed bacteria cells and biofilms on MFC anodes were stained with LIVE/DEAD™ BacLight™ Bacterial Viability Kit

(L7012, ThermoFisher, USA) using the manufacturer protocol. The stained cells were observed under CLSM (Zeiss 700). This Kit was also used to evaluate the number and viability of GY32 cells in sterilized sediments. Before being stained, 1 g wet sediment from each SMFCs was diluted in 10 mL sterilized PBS. Totally, 1 mL of the diluted sediment suspensions were then stained with the Bacterial Viability Kit and observed under CLSM. The numbers of filamentous cells (cell length ≥ 20 μm) were counted. Bacteria cell clusters in MFCs and the nanowire appendage product were stained with Nano-Orange (N10271, ThermoFisher), a protein-specific fluorescent dye[24,25], and then observed under CLSM as described by the manufacturer protocol.

**TEM**. TEM observation of *L. varians* GY32 cells and cell sections was conducted as our previous report[18]. After being cultivated aerobically for 12 h in LB, *L. varians* GY32 cells and cellular envelope structures were observed with TEM (HITACHI, H-7650). PBS-washed bacterial cells were used to prepare the section samples. The washed cells were fixed with glutaraldehyde (G7651, Sigma-Aldrich, USA) over-night, and then washed with PBS for five times, followed by osmium tetraoxide (75632, Sigma-Aldrich, USA) fixation for 40 min and washing again. Sections of 0.5–1 mm$^3$ were dehydrated with ethanol and acetone, followed by epoxide resin and sectioning. Section samples were also observed with TEM after being stained by uranyl acetate and lead acetate. The TEM investigation was performed for four independent cultures of *L. varians* GY32 all showing similar results.

**Transcriptome analysis**. GY32 cells were collected from MFCs containing 5 mM of acetate as the electron donor and from a control culture of GY32 cells grown aerobically in 30 mL of the same medium. Each cultivation was performed in tri-plicate. All cells were harvested in the middle of the exponential growth phase for transcriptome analysis by centrifugation (5000g, 10 min). Cell pellets were resus-pended in 1 mL of RNA protect bacterial reagent (76506, Qiagen, USA) for 5 min, collected by centrifugation (5000g, 10 min), and then resuspended in 200 μL of RNase-free water. RNA from the samples was extracted using an RNeasy kit (74104, Qiagen, USA) and purified with RNase-free DNase I (79254, Qiagen, USA) to digest residual genomic DNA. RNA quality was verified using a bioanalyzer (model 2100, Agilent Technologies). Equal amounts of the six RNA samples were used for further analysis. rRNA was removed by using a RiboZero rRNA removal kit (RZH1046, Epicenter, USA) according to the manufacturer's protocol. The resulting mRNA was used to construct a 300–594 bp cDNA library and sequenced on a HiSeq 2500 platform (Illumina, USA) in the same lane. Raw sequence data were deposited at the Gene Expression Omnibus (https://www.ncbi.nlm.nih.gov/geo/) with accession number GSE165753. Raw reads were checked and visualized with FastQC (Version 0.11.2, http://www.bioinformatics.babraham.ac.uk/projects/fastqc/). Clean reads were aligned with the reference genome using Bowtie2 tool (version 2.3.2) to identify known genes and to calculate gene expression by RNA-Seq Expectation-Maximization. The gene expression level was further normalized by using the transcripts per kilobase million mapped reads method to eliminate the influence of different gene lengths and amount of sequencing data on the calculation of gene expression. The edge R package (http://www.r-project.org/) was used to identify differentially expressed genes across samples with fold changes ≥ 2 and a false discovery rate adjusted *P* (*q* value) < 0.05. DEGs were then subjected to enrichment analysis of gene ontology (http://geneontology.org/) function and Kyoto encyclo-pedia of genes and genomes (https://www.genome.jp/kegg/pathway.html) pathways, and *q* values were corrected using <0.05 as the threshold.

**Reporting summary**. Further information on research design is available in the Nature Research Reporting Summary linked to this article.

## Data availability
The transcriptome data that support the findings of this study have been deposited in the Gene Expression Omnibus with accession number GSE165753. Source data are provided with this paper.

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

## Acknowledgements

We thank Dr. Lars Riis Damgaard from Aarhus University for his help on this work. This study was supported by the National Natural Science Foundation of China (91851202, 31970110 and 52002254); Guangdong Provincial Science and Technology Project (2016A030306021, 2020B1111380003); Science and Technology Project of Guangdong Academy of Sciences (2019GDASYL-0301002, 2020GDASYL-20200402001); Open Project of State Key Laboratory of Applied Microbiology Southern China (SKLAM001-2018), Sichuan Science and Technology Program (2020YJ0262), Chunhui plan of Ministry of Education of China, the Sino-Danish Centre for Education and Research, Danish Council for Independent Research (grant no. 6108-00396B), Danish National Research Foundation (DNRF136), EU H2020 Marie Sklodowska-Curie Actions (MNR4SCELL no. 734174, SENTINEL no. 812398), the Carlsberg Foundation, Fundamental Research Funds for the Central Universities, China (YJ201893), State Key Lab of Advanced Metals and Materials, China (Grant No. 2019-Z03) and Open Project of State Key Laboratory of Applied Microbiology Southern China (SKLAMM001-2018).

## Author contributions

M.X., L.P.N., and M.D. conceived and supervised the project. Z.W., Y.Y., R.B., M.M., and J.M. carried out experiments including nanowire and bacteria cell conductivity tests. Y.Y., G.S., J.G., and Y.M. carried out the bacteria cultivation and CLSM observation. L.H.K., C.G., and D.L. conducted the AFM measurements. Y.Y., G.K., D.L., J.T.B., and C.G. performed MFC and MES experiments and data analysis. G.K. and C.Z. analyzed the genome sequence and gene expression data. All authors discussed the results. Y.Y., L.H.K., and M.D. performed data analysis and wrote the paper with comments from all the other authors.

## Competing interests

The authors declare no competing interests.
