## [Peer Review File · Nature Communications]

REVIEWER COMMENTS

Reviewer #1 (Remarks to the Author):

In this work, Mingdong Dong and co-workers demonstrated long-range extracellular electron transfer (EET) by *Lysinibacillus varians* GY32 under anaerobic conditions. There is a widespread interest in the role and mechanisms of EET in electroactive bacteria and the current research makes a significant contribution through a good combination of microscopic and electrochemical studies.

The manuscript is clearly presented and well-written and I recommend this work for publication in Nature Communications. I included some minor suggestions below.

Statement that the cell wall of Gram-positive bacteria is non-electron conductive is not completely correct (page 4). Peptidoglycan, lipids and lipopolysaccharides composing cell walls are non-conductive indeed. However, it is a complex and dynamic structure, there is a number of cell wall associated, covalently and non-covalently bound, proteins, which can participate in electron transfer across the cell envelope. For instance, flavin-based proteins associated with the cell surface were demonstrated to be involved in EET processes for *Listeria* (Nature 562, 140–144, 2018) and confirmed for *Enterococcus* (J. Bacteriol. 202: e00725-19).

In contrast to Gram-negative strains, the cell envelope associated multiheme c-type cytochromes are a very uncommon property for Gram-positive bacteria. Until now there are only two reported cases of involvement of those proteins in EET processes: *Thermincola potens* (relying on the cell wall associated c-cytochromes) and *Carboxydotherrmus ferrireducens* (carrying c-cytochromes in the S-layer, Geomicrobiol. J. 29, 804-819, 2012). The latest one should be cited in the work too. Are the presented current densities mean values? Standard deviation should be reported too. How many electrodes were tested in each experiment?

Reviewer #2 (Remarks to the Author):

This is a very clear, well-written report of the first discovery of a gram+ bacteria that appears to utilize conductive filaments for long-distance EET. I think my colleagues who work in microbial electrochemistry will find it interesting - but I am on the fence as to whether it merits publication in Nature Communications. It is not the first discovery of bacterial conducting filaments - just the first for gram+'s. so I challenge to authors to make a stronger case as to why this is so noteworthy - why some may have thought not possible, etc. A few brief technical notes: 1) not being able to detect H₂ does not necessarily mean it is not being generated - the cells may consume it as fast as the electrode makes it. 2) I'm surprised there is no voltammetry which may reveal redox features. 3) With regards to the conductivity measurements, relevant models predict that conductivity will be dependent on the charge state of the material. I would suggest performing an electrochemical gate measurement whereby both electrodes are independently biased vs. a ref, start with both at an oxidizing potential, then sweep one cathodically to create the bias to derive electron transport. Throughout the measurement the charge state will change and the dependency of current (conductivity) on potential would be very informative as to the mechanism.

Reviewer #3 (Remarks to the Author):

The authors of this manuscript report how *Lysinibacillus varians* GY32, a filamentous, unicellular, Gram-positive bacterium can transfer electrons bidirectionally. Atomic Force Microscopy (AFM) and microelectrodes were used to show that the conductivity is linked to pili-like protein nanowire appendages that could reach a length of approximately 1 mm when grown on graphite electrodes. Bacterial long-distance electron transfer (LDET) is an important, recently discovered process; therefore, I believe this manuscript is timely and focuses on an important topic.

While the title interested me, I have several fundamental issues with the manuscript. My primary issue is the way the authors wrote the methodology. I believe published papers ought to have a more in-depth description of the methodology than was presented here.

Electron transfer by the electrode grown bacteria is controlled by the electrode potentials. This study uses MFC as its mode of operation. There is a flaw in this method, as knowing and controlling the potential of each electrode is critical for reproducibility. The authors' cathode potential (no methodology given) was -0.6 V (Fig 2 D). MFC mode was used for Fig 2A. If the electron transfer is bidirectional, I would expect both measurements to be taken using identical mode of operation (polarized electrodes). The MFC mode shows a redox gradient in the experimental system, a source of current flow that does not illustrate bidirectional current.

The methodology needs to be re-written in such detail that others can replicate the work. For example, the authors state that, "The Fe²⁺ concentration was tested by ferrozine assay and H₂ concentration at the surface of cathode was monitored with a H₂ microelectrode (Unisense)." Which ferrozine assay and H₂ microelectrodes were used? Please provide references and catalog numbers.

"Conductivity tests and calculation" The authors claim mm level electron transfer, however, this process is not discussed in the protocol. How was electron transfer measured at the mm level? In Fig. 4, the authors use 20 μm gap distances, yet it is clear that there are more than single cells on the gap which contribute to electron transfer (Fig. 4E). The reported conductivity values are incongruous with the 100s of data reported in Fig. 4F. The authors claim 10 days of conductivity but present 100 s data. It is unclear how this can be 10 days, without extending the measurements several hours.

Fig. 2: Please add biological replicates for B and C (average and SD).

Fig. 2E: demonstrates that GY32 catalyse oxygen reduction. Do the authors have O₂ profiles for the same conditions?

Fig. 2F: Even at time zero, there was H₂ in the system.

Line 74: Replace "relative" with "relatively."

Please provide the source and catalog number of each chemical used for this study. (i.e. 10 mM of formate or acetate). These may be critical for others who want to replicate the study.

Lines 154-155: What happens after 10 days? The authors should provide 10 days of data (with biological replicates) in the manuscript.

Line 302: "The percentage of aromatic amino acids in GY32-ComGD is 13.8%" What is error here? Please include error for each reported value in the manuscript.

Fig. 5: The authors have not reported conductivity measurements and its characterization (similar to Lovly and Reguara papers) but claim that *L. varians* transfer electrons using nanowires.

How was the system operated anaerobically? This is a critical component of the experimental system but no description was included.

REVIEWER COMMENTS

Reviewer #1 (Remarks to the Author):

In this work, Mingdong Dong and co-workers demonstrated long-range extracellular electron transfer (EET) by *Lysinibacillus varians* GY32 under anaerobic conditions. There is a widespread interest in the role and mechanisms of EET in electroactive bacteria and the current research makes a significant contribution through a good combination of microscopic and electrochemical studies. The manuscript is clearly presented and well-written and I recommend this work for publication in Nature Communications. I included some minor suggestions below.

Re: Thank you very much for your nice comments and suggestions. We have revised the manuscript according to the comments of you and the other reviewers. All revised text in the manuscript has been highlighted in red. We believe the manuscript has been significantly improved with your comments.

Statement that the cell wall of Gram-positive bacteria is non-electron conductive is not completely correct (page 4). Peptidoglycan, lipids and lipopolysaccharides composing cell walls are non-conductive indeed. However, it is a complex and dynamic structure, there is a number of cell wall associated, covalently and non-covalently bound, proteins, which can participate in electron transfer across the cell envelope. For instance, flavin-based proteins associated with the cell surface were demonstrated to be involved in EET processes for *Listeria* (Nature 562, 140–144, 2018) and confirmed for *Enterococcus* (J. Bacteriol. 202: e00725-19).

Re: Thank you for pointing out this inaccuracy. When describing the structure of *L. varians* GY32 as observed by TEM we wanted to emphasize the unexpectedness of conductivity in Gram-positive bacteria as compared to Gram-negative bacteria. We did not intend to indicate that the cell wall of Gram-positive bacteria is exclusively non-electron conductive, and we have therefore revised the paragraph to better reflect this:

“the thick cell wall is generally considered to be a limiting factor for the extracellular electron shuttling capacity of Gram-positive bacteria (19).” (revised manuscript line 123-125).

We have further added a description of the interesting instance of EET in Gram-positive bacteria mentioned by the referee:

*“EET strategies generally involve c-type cytochromes or electron shuttles, and EET has been observed for the Gram-positive bacteria *Enterococcus faecalis* and *Listeria monocytogenes* using flavins as electron shuttles (16,30-33).”* (revised manuscript line 295-298).

In contrast to Gram-negative strains, the cell envelope associated multiheme c-type cytochromes are a very uncommon property for Gram-positive bacteria. Until now there are only two reported cases of involvement of those proteins in EET processes: *Thermincola potens* (relying on the cell wall associated c-cytochromes) and *Carboxydotherrmus ferrireducens*

(carrying c-cytochromes in the S-layer, Geomicrobiol. J. 29, 804-819, 2012). The latest one should be cited in the work too.

Are the presented current densities mean values? Standard deviation should be reported too. How many electrodes were tested in each experiment?

Re: Thank you for sharing this insight. We have expanded the discussion on EET in Gram-positive bacteria, including the mentioned articles:

“Gram-positive bacteria are ubiquitous in various environments and can be dominant in some iron- or electrode-reducing microbial communities (19, 43). Although several of them have been demonstrated to perform EET via outer membrane cytochromes or electron mediators, current knowledge on the strategies of Gram-positive bacteria to participate in microbial EET and LDET networks are much less compared to these of Gram-negative bacteria (3, 19, 31-34).” (revised manuscript line 361-366).

The presented current densities were previously mean or typical current densities for the microbial electrochemical tests. In the revised manuscript, average values and standard deviations have been provided. The electrode numbers for each experiment were at least three. All the numbers of replicates in each experiment have been reported throughout the revised manuscript.

Reviewer #2 (Remarks to the Author):

This is a very clear, well-written report of the first discovery of a gram+ bacteria that appears to utilize conductive filaments for long-distance EET. I think my colleagues who work in microbial electrochemistry will find it interesting - but I am on the fence as to whether it merits publication in Nature Communications. It is not the first discovery of bacterial conducting filaments - just the first for gram+'s. so I challenge to authors to make a stronger case as to why this is so noteworthy - why some may have thought not possible, etc.

Re: Thank you very much for your comments and suggestions. We have revised the manuscript according to the comments of you and the other reviewers. All revised text have been highlighted in red. We believe the manuscript has been significantly improved with your comments.

We believe our results merit publication in Nature Communication for the following reasons/novelities:

(1) Longest Unicellular bacterium

Here we reported the longest unicellular bacterium known to date, which will be of wide interests to microbiologists. Bacterial shape affects critical biological functions (e.g. nutrient acquisition, motility, stress resistance and interactions with other organisms) and thus bacterial morphology is always a key field in microbiology (Nature 1981, 290:797–799; Nature, 2014, 506: 489–493; Nat Rev Microbiol 2015, 13: 241–248; Cell, 2017, 168: 172-185; Science 2017, 355(6326):739-743). *L. varians* GY32 provides a unique model for microbial morphology

studies, which will expand our knowledge on bacterial shape and inspire novel mechanisms on bacterial cell division, environmental adaptation and functions.

(2) First conductive nanowires from Gram-positive bacteria:

We reported the first conductive nanowires and cellular networks generated by Gram-positive bacteria. Gram-positive bacteria are ubiquitous or even dominant in iron- or electrode-reducing microbial communities (Kunapuli et al., ISME J, 2007, 1: 643–653; Wrighton et al., ISME J, 2008, 2: 1146–1156). Although the EET pathways of Gram-positive bacteria have been paid wide attentions (Light et al., Nature, 2018, 562: 140–144; Carlson et al., PNAS, 2012, 109: 1702-1707), our knowledge on the EET strategies of Gram-positive bacteria are much less compared to Gram-negative bacteria. Our finding of conductive nanowires of Gram-positive bacteria is an essential part to complete the whole picture of the microbial EET and LDET networks in nature environments. As commented by the other reviewers: “There is a widespread interest in the role and mechanisms of EET in electroactive bacteria and the current research makes a significant contribution through a good combination of microscopic and electrochemical studies.” ; and “I believe this manuscript is timely and focuses on an important topic.”

(3) Novel type of LDET network

The filamentous cells and extracellular nanowires of GY32 co-formed a novel type of microbial LDET networks. Two types of microbial LDET networks have been intensively studied and both were firstly published in Nature (Reguera et al., Nature, 2005, 435: 1098–1101; Pfeffer et al., Nature, 2012, 491: 218-221). (i) Short cell integrated with extracellular nanowires, typically anaerobic bacterium *Geobacter*. Such LDET networks depend largely on micrometer-scale biofilms formed on particle surfaces. (ii) Filamentous cable bacteria with serially connected cells and intracellular nanowires. This LDET network can span centimeter distance but was mainly observed at sediment-water interfaces. LDET network of gram-positive GY32 comprises filamentous, unicellular cell and extracellular nanowires, like a combination of the two reported LDET network modes of Gram-negative bacteria. Centimeter long cell networks of GY32 can form around the solid electron acceptors (Fig. 3A) and may exist in both oxic and anoxic environments. Besides, one single GY32 cell, inside or outside the networks, may connect environments or organisms separated by millimeter-distance and connect them by simultaneous communications of signals, electrons and metabolites.

Although how GY32 networks function in real environments and the underlying mechanisms are unrealized, the current results will inspire and lead to many novel and important knowledge in the fields of microbiology and environmental sciences. Therefore, we believe that this report merits the publication in Nature Communication and will arise wide interests of the readership. We have now expanded the description of EET in Gram-positive bacteria and LDET findings in Gram-negative bacteria to highlight the novelty of GY32 throughout the manuscript, and we have expanded the discussion describing the mentioned 3 points in the revised manuscript:

*“Microbial cell shape affects critical biological and ecological functions. The unique cellular shape of *L. varians* GY32 indicates special strategies underlying its cell division, nutrient acquisition, motility, environmental sensing and interactions with other organisms.”*

“Gram-positive bacteria are ubiquitous in various environments and can be dominant in some

iron- or electrode-reducing microbial communities^{16,43}. Although several of them have been demonstrated to perform EET via outer membrane cytochromes or electron mediators, current knowledge on the strategies of Gram-positive bacteria to participate in microbial EET and LDET networks are much less compared to these of Gram-negative bacteria^{3,16,31-34}.”

“More and more evidences suggests a wide distribution of microbial LDET in natural and engineering environments^{3-5,7,11,28}. Two types of microbial LDET network have been intensively studied. Firstly, short cells (e.g. *Geobacter* and *Shewanella*) integrated with extracellular nanowires. These LDET networks depend largely on the biofilm formation on particle surfaces and have a limited thickness (generally ≤ 0.1 mm)³⁸. Secondly, filamentous cable bacteria with serially connected cells and intracellular nanowires^{7,8}. Cable bacteria LDET networks can span centimetre distances but exist mainly at sediment-water interfaces. The LDET networks of gram-positive *L. varians* GY32 comprise filamentous, unicellular cells and extracellular nanowires, like a combination of the two LDET network modes of Gram-negative bacteria (Fig. 5). Centimetre long cell networks of GY32 can form around the solid electron acceptors (Fig. 3A). Moreover, the networks of this facultative anaerobic bacteria may exist in both oxic and anoxic environments, vastly increasing the area where LDET plays a role.”

A few brief technical notes: 1) not being able to detect H₂ does not necessarily mean it is not being generated - the cells may consume it as fast as the electrode makes it.

Re: The referee raises an interesting point. Not being able to detect H₂ is not necessarily sufficient for concluding that it is not being generated. We tested the capacity of *L. varians* GY32 for consuming H₂ as electron donor to reduce Fe(III), and found that *L. varians* GY32 could not use H₂ for Fe(III) reduction within 48 h, while *Shewanella oneidensis* MR-1, as a positive control, showed significant Fe reduction within 2 h. Considering that *L. varians* GY32 does not consume H₂, we believe that generated H₂ would not be so rapidly consumed that it could not be detected. This result has been added in the revised manuscript line 182-184 and Fig. S3:

Fig. S3 *L. varians* GY32 cannot use H₂ to reduce Fe(III) (n = 3). *L. varians* GY32 and *Shewanella oneidensis* MR-1 were inoculated in sterilized PBS buffer (pH 7.0, initial OD600 = 0.1) containing 1 mM of ferric citrate in serum bottles. The buffer was firstly bubbled with nitrogen to remove oxygen (verified by O₂ microelectrode, H₂-25-7343, Unisense) and then bubbled with pure hydrogen until the H₂ concentration reached 0.5 mM (measured by H₂ microelectrode,

H2-25-7343, Unisense). The serum bottles were sealed with butyl rubber and aluminium caps. S. oneidensis MR-1 was used as a positive control as it has the capability to use H₂ as electron donor to reduce ferric citrate (Liu et al., 2002, Biotechnol Bioeng 80: 637–648).

2) I'm surprised there is no voltammetry which may reveal redox features.

Re: We agree with the reviewer that cyclic voltammetry is interesting for testing the electrochemical activity of GY32 biofilms. We have now performed cyclic voltammetry tests under turnover and non-turnover conditions, (revised manuscript line 171-176, Figure S2G). The CV profile of GY32 biofilms showed a pair of redox peaks (an oxidative peak at 0.09 V and a reductive peak at -0.08 V vs SHE) under both conditions, which was consistent with the reversible electron transfer between GY32 and electrode:

“Cyclic voltammetry curves of anodic GY32 biofilms showed an oxidative peak at 0.09 V and a reductive peak at -0.08 V vs SHE (Fig. S2G) under non-turnover condition. Similar cyclic voltammetry profile with higher peak-current was observed in the presence of acetate as electron donor (turnover condition) indicating redox species generated within biofilms.” (revised manuscript line 170-174)

Fig. S2 (G) Representative cyclic voltammetry curves of anodic GY32 biofilms under turnover and non-turnover conditions.

3) With regards to the conductivity measurements, relevant models predict that conductivity will be dependent on the charge state of the material. I would suggest performing an electrochemical gate measurement whereby both electrodes are independently biased vs. a ref, start with both at an oxidizing potential, then sweep one cathodically to create the bias to derive electron transport. Throughout the measurement the charge state will change and the dependency of current (conductivity) on potential would be very informative as to the mechanism.

Re: Thank you for your suggestions. We have conducted electrochemical gate measurements with *L. varians* GY32 biofilms grown on interdigitated microelectrode arrays (IMAs), according to the methods reported by Yates et al. (Nat Nanotechnol, 2016, 11: 910-913). The results showed a peak conductivity of 0.25 ± 0.1 mS/cm at 0.05 V (gate potential E_g , vs SHE) when the gate

potential (E_{gate}) increased from -0.4 to 0.2 V (vs SHE), which suggested that the conductivity of *L. varians* GY32 cellular networks is a redox process. The results have been provided in line 235-239 and Fig. S6:

Fig. S6 Conductivity of GY32 biofilms based on electrochemical gating measurements. The dotted curves indicate conductivities of three biofilm samples and the dashed line indicates the averaged conductivities of the three samples (E_{gate} : gate potential). Insert shows the GY32 biofilms on IMAs. The image with a white square shows a zoom-out view of the biofilms in which the dark-purple areas indicate dense biofilms and light-purple areas indicate rare biofilms. The white arrow indicates a zoom-in of the white square and the dark-purple filaments indicate GY32 cells.

Reviewer #3 (Remarks to the Author):

The authors of this manuscript report how *Lysinibacillus varians* GY32, a filamentous, unicellular, Gram-positive bacterium can transfer electrons bidirectionally. Atomic Force Microscopy (AFM) and microelectrodes were used to show that the conductivity is linked to pili-like protein nanowire appendages that could reach a length of approximately 1 mm when grown on graphite electrodes. Bacterial long-distance electron transfer (LDET) is an important, recently discovered process; therefore, I believe this manuscript is timely and focuses on an important topic.

While the title interested me, I have several fundamental issues with the manuscript. My primary issue is the way the authors wrote the methodology. I believe published papers ought

to have a more in-depth description of the methodology than was presented here.

Re: Thank you very much for your comments and suggestions. We have revised the manuscript according to the comments of you and the other reviewers. We apologize for the lacking Methods section, it has now been thoroughly revised and a systematic investigation of MFC electrode potentials has been performed. We have also updated the schematics in all figures throughout the manuscript to better illustrate the methodology, especially Fig.2 should now much better explain the experimental setup:

Fig. 2. Bidirectional EET between GY32 and electrodes. (A) Schematic of GY32 electricity generation in SMFCs. (B) Electricity generation by inoculating GY32 in SMFCs. (C) Schematic of GY32 EET in MFCs with liquid medium (upper reactor) and in BCESs (lower reactor). (D) Electricity generation by GY32 in MFCs with liquid medium. (E) Fe(III) reduction by GY32 with a cathode (-0.6 V vs SHE) as the sole electron donor in BCESs. (F) H₂ concentration in the BCESs with or without GY32, insert shows the GY32 biofilms on cathode. *n* = 3.

All revised text in the manuscript has been highlighted in red. We believe the manuscript has been significantly improved with your comments.

Electron transfer by the electrode grown bacteria is controlled by the electrode potentials. This study uses MFC as its mode of operation. There is a flaw in this method, as knowing and controlling the potential of each electrode is critical for reproducibility. The authors' cathode potential (no methodology given) was -0.6 V (Fig 2 D). MFC mode was used for Fig 2A. If the electron transfer is bidirectional, I would expect both measurements to be taken using identical mode of operation (polarized electrodes). The MFC mode shows a redox gradient in the experimental system, a source of current flow that does not illustrate bidirectional current.

Re: Thank you for pointing out this possible discrepancy. The schematics in the original Fig. 2A+D were not clearly illustrating the experimental setup. We have now updated the schematics (see Fig. 2A+C in the reply to the previous comment). The cathode potential in the bio-cathode electrochemical systems (BCESs) was poised at -0.6 V (vs standard hydrogen electrode) by using an electrochemical workstation (CHI 1040C, China) and an Ag/AgCl reference electrode. We hope that this is now clearer in the schematic in Fig. 2C (previously Fig. 2D).

To perform a more systematic investigation of the electrode polarization effect, we have followed your suggestions and tested the current generation by GY32 with anodes polarized at different potentials including -0.2, 0, 0.2, 0.4, 0.6 V, vs standard hydrogen electrode (SHE). GY32 showed the highest current density when using a 0.4 V anode as electron acceptor. The results of GY32 using acetate and formate to reducing 0.4 V-polarized anodes have been provided in the revised manuscript (line 166-170, Fig. S2F). The detailed methodology of GY32 using polarized electrodes as electron acceptors or electron donors in bioelectrochemical systems has been provided in the revised Materials and Method section (line 436-442; line 455-488):

*“To further test the EET capability of *L. varians* GY32 to a polarized electrode, the same MFC reactors and electrolyte were prepared except for that the external resistors were replaced with an electrochemical workstation (CHI 1040C, China) and an Ag/AgCl reference electrode (1038, Gaoss Union, China) was placed in each anode chamber. The anodes were polarized at 0.203 V vs Ag/AgCl reference electrodes (i.e. 0.4 V vs SHE). The current between anode and cathode was recorded for every 100 seconds under 30 °C by the electrochemical workstation.”*

Fig. S2F Current generation by strain GY32 using a polarized anode (0.4 V vs standard hydrogen electrode) as electron acceptor and acetate or formate as electron donor.

The methodology needs to be re-written in such detail that others can replicate the work. For example, the authors state that, “The Fe²⁺ concentration was tested by ferrozine assay and H₂ concentration at the surface of cathode was monitored with a H₂ microelectrode (Unisense).” Which ferrozine assay and H₂ microelectrodes were used? Please provide references and catalog numbers.

Re: More details of the methodology and catalog numbers of each chemical or material used in this study have been provided to make sure that others can replicate (see the whole Materials and Methods section, line 398-586). The reference of ferrozine assay has been provided (Ref. 46, line 475). And catalog number (H2-25, 7343, Unisense, Denmark) of the H₂ microelectrode has also been provided (line 476).

“Conductivity tests and calculation” The authors claim mm level electron transfer, however, this process is not discussed in the protocol. How was electron transfer measured at the mm level? In Fig. 4, the authors use 20 μm gap distances, yet it is clear that there are more than single cells on the gap which contribute to electron transfer (Fig. 4E). The reported conductivity values are incongruous with the 100s of data reported in Fig. 4F. The authors claim 10 days of conductivity but present 100 s data. It is unclear how this can be 10 days, without extending the measurements several hours.

Re: Thank you for your comments. Indeed, we claimed “*The electron delivery distance of a single GY32 cell appears to be on orders of magnitude shorter than that of cable bacteria.*” (line 370-371 in the original manuscript). Although the maximum length of a single GY32 cell can be over 1 mm (Fig.1 and Fig. S1), we did not test the electron transfer along such a filamentous cell. This claim has been removed in the revised manuscript.

On Fig.4, Fig.4ABC shows the conductivity test of single GY32 cell bridging two fabricated electrodes. Although sometimes there were more than one cells in the view, we only tested two electrodes that were connected by one single cell. The other cells did not connect these two electrodes and will not contribute to the current signal. To show the single-cell tests more clearly, Fig 4B has been replaced with a more typical figure so that concerns of multiple cells can be excluded. The filaments shown in Fig.4E are extracted nanowires rather than bacteria cells on electrode arrays, which should now be clearly described in Fig.4D+E and the figure text:

Fig.4. Conductivity measurements of GY32 cell envelope and nanowires by microelectrodes. (A) Schematic of the single cell measurement; (B) Single cell on an electrode array observed under light microscope; (C) Current-voltage curve of a single GY32 cell ($n = 5$); (D) Schematic of the nanowire measurement; (E) nanowire network on an electrode array observed by AFM, dark line indicates a 100 nm height different between the gap and electrode; (F) Current-time curve of the hydrated nanowire products (the voltage is 0.1 V, $n = 8$).

Concerning the claim of 10 days of conductivity, we described the current generation of GY32-SMFCs with “Stable current could be maintained for over ten days” (line 154 in original manuscript) but not the conductivity of GY32 cells or nanowires. We typically tested the conductivity of cell clusters on IMAs or fabricated electrode arrays for 200 s, but for this experiment 1-2 μl of cells or nanowire appendages washed in deionized water were tested. Under these conditions we do not expect stable conductivity for extended time periods. We have expanded the methodology to better avoid misunderstandings:

“The same IMAs used in above electrochemical gate measurements were also used to test the conductivity of nanowire appendages extracted from GY32 culture. 1 μl buffer containing nanowire appendages ($n = 8$), or ethanolamine buffer without nanowire appendages as a control, was loaded on an electrode array. The electrode array was then observed with AFM to confirm the connection of electrodes by nanowires (Fig. 4E). The I-V (from -0.1 to 0.1 V, scan rate 1 mV/s) and I-t (0.1 V maintained for 200 seconds) profiles of the electrode array connected by nanowires

were tested.

To test the conductivity of the individual cells, 2 μL of GY32 culture, after being diluted by 1000 fold with deionized water, were loaded on a laboratory fabricated electrode array (Fig. 4B). Cells ($n = 5$) connecting two or more electrodes were located under optical microscope and the I-V profile between the electrodes connected by these cells were tested (from -0.1 to 0.1 V, scan rate 2 mV/s) with the probe station equipped with a Keithley 2614B sourcemeter in ambient environment (25 $^{\circ}\text{C}$).” (line 554-566)

Fig. 2: Please add biological replicates for B and C (average and SD).

Re: The results presented in Fig. 2 B+C were previously representative curves. We have now included a minimum of three biological replicates for all presented results, and the data in Fig. 2 B+D (previously Fig. 2 B+C) is now presented as the average and standard deviation from three biological replicates. Please note that the two curves have changed slightly due to the inclusion of additional data, but the conclusions remain unchanged:

Fig. 2E: demonstrates that GY32 catalyse oxygen reduction. Do the authors have O2 profiles for the same conditions?

Re: The methodology was not clear in the original manuscript. In Fig. 2E we showed the capacity of *L. varians* GY32 for reducing Fe(III) using a cathode as sole electron donor in a biocathode electrochemical system. In this experiment we depleted the oxygen by bubbling the chamber with pure N_2 until the dissolved oxygen in the medium was below 0.3 μM as measured with an oxygen microelectrode (detection limit: 0.3 μM , OX-14125, Unisense, Denmark):

“The cathode chambers were bubbled with pure N_2 until the oxygen in the PBS was below 0.3 μM as measured with an oxygen microelectrode (detection limit: 0.3 μM , OX-14125, Unisense, Denmark) and then sealed with butyl rubbers and plastic caps.” (method section, line 470-474)

We have furthermore tested the oxygen reduction of GY32 by performing an experiment where the chamber was bubbled with air until the dissolved oxygen was elevated to $233.8 \pm 2.7 \mu\text{M}$:

“Similarly, three dual-chamber BCEs were assembled for cathodic oxygen reduction. The catholyte was 100 mL sterilized PBS and the dissolved oxygen was elevated to $233.8 \pm 2.7 \mu\text{M}$ by bubbling with air. The initial OD600 of PBS-washed GY32 cells in catholyte was 0.05. Each anode

chamber was supplemented with 100 mL sterilized PBS containing 50 mM of potassium ferricyanide. The cathode potential was poised at 0.1 V (vs SHE) by using an electrochemical workstation (CHI 1040C, China) and an Ag/AgCl reference electrode. The cathode chambers were sealed with butyl rubber gaskets and plastic caps. BCESs with either unpolarised cathodes or GY32-free catholyte were operated as controls. The dissolved oxygen in the cathode chamber was tested using the oxygen microelectrode (OX-14125, Unisense, Denmark). All BCESs were tested in an anaerobic workstations (AEP, Electrotek, United Kingdom) under 30 °C. The setup is illustrated in Fig. 2C.” (method section line 476-488).

The results demonstrates that GY32 could use cathode as electron donor to reduce oxygen using electrodes polarized at 0.1 V (vs SHE). In the BCES with polarized electrode and GY32, the oxygen reducing rate ($2.5 \pm 0.3 \mu\text{M/h}$) was much higher than the BCES with only GY32 ($0.9 \pm 0.2 \mu\text{M/h}$) or polarized electrodes ($0.3 \pm 0.1 \mu\text{M/h}$). The oxygen reduction by PBS-washed GY32 without a polarized electrode may be attributed to the inherent nutrients in the cells. The result has been described in line 185-187, the current and oxygen reduction profiles have been provided in Fig. S4:

Fig. S4 O₂ reduction profile of L. varians GY32 using electrode polarized at 0.1 V (vs standard hydrogen electrode) as electron donor. In the BCES with polarized electrode and GY32, the oxygen reducing rate ($2.5 \pm 0.3 \mu\text{M/h}$) was much higher than the BCES with only GY32 ($0.9 \pm 0.2 \mu\text{M/h}$) or polarized electrodes ($0.3 \pm 0.1 \mu\text{M/h}$) (n = 3). The oxygen reduction by PBS-washed GY32 without a polarized electrode may be attributed to the inherent nutrients stored within the cells.

Fig. 2F: Even at time zero, there was H₂ in the system.

Re: Fig. 2F showed that the concentration of H₂ in both reactors with or without GY32 were very low (<0.1 µM) from time zero to the end. The detection limit of the H₂ microelectrode is 0.1 µM in water (<https://www.unisense.com/H2/>) which means no detectable amounts of H₂ was accumulated throughout the experiment.

Line 74: Replace “relative” with “relatively.”

Re: We have replace “relative” with “relatively” in line 73 in the revised manuscript.

Please provide the source and catalog number of each chemical used for this study. (i.e. 10 mM of formate or acetate). These may be critical for others who want to replicate the study.

Re: Sources and catalog numbers of all chemicals used in this study have been provided in the revised manuscript (Materials and Methods, line 398-586).

Lines 154-155: What happens after 10 days? The authors should provide 10 days of data (with biological replicates) in the manuscript.

Re: Current generation by sediment microbial fuel cells can last for several months or years. Generally, the current increases to the maximal value (generally within ten days for SMFCs < 0.5 L, Yates et al., ISME J, 2012, 6: 2002–2013; Wang et al., Chem Eng J, 2021, 405: 126680) and maintains for a certain time and then decreases to their background current due the consumption of electron donors and the inherent potential difference between sediments and the overlying water. Like the other types of MFCs, the relatively stable maximum current density was the key factor to evaluate the EET-activities of the microorganisms in the sediments.

We measured the current generating of SMFCs (with three biological replicates) for 20 days and the averaged current densities with SD have been provided in the revised manuscript (Fig. 2B). The relatively stable maximum current of GY32-innoculated SMFCs was increased by $75.2 \pm 7.1\%$ compared to non-innoculated SMFCs, demonstrating the contribution of GY32 to the current generation in the SMFCs. We believe the current of all SMFCs in our study will decrease after their relatively stable maximum current but that will not affect our conclusion that the presence of GY32 can increase the EET to electrodes in sediments.

Line 302: “The percentage of aromatic amino acids in GY32-ComGD is 13.8%” What is error here? Please include error for each reported value in the manuscript.

Re: ComGD is a single protein encoded by gene *comGD* (for example the ComGD of *Bacillus licheniformis*, NCBI Accession number: SPU11769). Therefore, there should be no error here (line 309 in the revised manuscript). We have included errors for all reported values in the revised manuscript.

Fig. 5: The authors have not reported conductivity measurements and its characterization (similar to Lovly and Reguera papers) but claim that *L. varians* transfer electrons using nanowires.

Re: The conductivity of *Geobacter* protein nanowires has been a topic of great interest for the last two decades. Reguera and Lovley initially used conductive-AFM (*Nature* 435.7045 (2005): 1098-1101) to indicate conductivity of these nanowires, and the results have later been supported by other methods, such as electrochemical gating and microelectrode array experiments (e.g. in the papers by Lovleys: *Energy & Environmental Science* 5.9 (2012): 8651-8659 and *RSC advances* 6.10 (2016): 8354-8357).

We used AFM and microelectrode arrays to study the conductivity of *L. varians* GY32 nanowires, and we have now added electrochemical gating experiments, which we believe provide a comprehensive and convincing picture of the nanowire conductivity:

- In our work we first used Kelvin Probe Force Microscopy (KPFM) to study the electrical properties of GY32 nanowires. KPFM reveals the sample electronic properties in the form of the surface potential, and we found a small potential difference between nanowire and gold that could indicate a much stronger conductivity of nanowires than the cell envelope of GY32 (Fig. 3F+G and revised manuscript line 239-255).
- We next isolated GY32 nanowires and deposited them on multi electrode arrays. The nanowires formed networks, which showed a strong conductivity (Fig. 4D-F and revised manuscript line 267-283). The formed network is essential for the conductivity in GY32 as we illustrate in Fig. 5. "*L. varians* GY32 combines a filamentous shape with extracellular nanowires to form conductive networks" (Line 339-340).
- We have now also performed electrochemical gating measurements of biofilms grown on interdigitated microelectrode arrays to further characterize the dependency of conductivity on the gate potential (i.e. biofilm redox status) to give insights into the conductive mechanisms of *L. varians* GY32 biofilms. We believe that the obtained results provide further evidence for the conductivity of the cell networks and nanowires of *L. varians* GY32 (Fig. S6 and revised manuscript line 235-239 + 499-522):

Fig. S6 Conductivity of GY32 biofilms based on electrochemical gating measurements. The dotted curves indicate conductivities of three biofilm samples and the dashed line indicates the averaged conductivities of the three samples (E_{gate} : gate potential). Insert shows the GY32 biofilms on IMAs. The image with a white square shows a zoom-out view of the biofilms in which the dark-purple areas indicate dense biofilms and light-purple areas indicate rare biofilms. The white arrow indicates a zoom-in of the white square and the dark-purple filaments indicate GY32 cells.

Fig.5 provides a comparison of the shared and unique properties of the conductive networks of *Geobacter*, cable bacteria and *L. varians* GY32. What we want to declare in this figure is that, like *Geobacter* and cable bacteria, *L. varians* GY32 can generate conductive nanowires and use them to form a conductive network. We did not mean to conclude that *L. varians* cells transfers electrons from inside to outside or between cells using nanowires as more evidence would be needed. The Fig.5 caption has been revised as:

“Fig.5. A comparison among the conductive networks of Geobacter, cable bacteria and L. varians GY32. Geobacter can form conductive networks (i.e. biofilms) comprising short unicellular bacteria and extracellular conductive protein nanowires (32). Filamentous multicellular cable bacteria conduct electricity through intercellular nanowires in the bacterial envelope (8). L. varians GY32 combines a filamentous shape with extracellular nanowires to form conductive networks.” (Line 335-340 in the revised manuscript)

How was the system operated anaerobically? This is a critical component of the experimental system but no description was included.

Re: Thank you. The anode/working chambers of the MFCs or BCESs were bubbled with pure N₂ until the oxygen in the medium was undetectable with an oxygen microelectrode (detection limit: 0.3 μM, OX-14125, Unisense, Denmark). These information has been provided Materials and Methods section in the revised manuscript (Line 423-426, line 462-464, line 477-478).

REVIEWERS' COMMENTS

Reviewer #2 (Remarks to the Author):

Authors have thoroughly addressed my concerns. Should be published in NC.

Reviewer #3 (Remarks to the Author):

Thanks for providing additional data and clarification. I believe they improved the manuscript.

REVIEWER COMMENTS

Reviewer #2 (Remarks to the Author):

Authors have thoroughly addressed my concerns. Should be published in NC.

Reply: We appreciate the reviewer's assistance in improving the manuscript.

Reviewer #3 (Remarks to the Author):

Thanks for providing additional data and clarification. I believe they improved the manuscript.

Reply: We appreciate the reviewer's assistance in improving the manuscript.